# County-level factors affecting Latino HIV disparities in the United States

**Nanette D. Benbow**[1]*, **David A. Aaby**[2], **Eli S. Rosenberg**[3], **C. Hendricks Brown**[1]

**1** Department of Psychiatry and Behavioral Sciences, Feinberg School of Medicine, Northwestern University, Chicago, Illinois, United States of America, **2** Department of Preventive Medicine, Feinberg School of Medicine, Northwestern University, Chicago, Illinois, United States of America, **3** Department of Epidemiology and Biostatistics, University at Albany School of Public Health, State University of New York, Rensselaer, New York, United States of America

* Nanette@northwestern.edu

**Data Availability Statement:** All data in this study are publicly available and can be found here: AIDSVu available at: https://aidsvu.org/resources/#/2016; County Health Rankings and Roadmaps available at: https://www.countyhealthrankings.org/explore-health-rankings/rankings-data-

## Abstract

### Objective

To determine which county-level social, economic, demographic, epidemiologic and access to care factors are associated with Latino/non-Latino White disparities in prevalence of diagnosed HIV infection.

### Methods and findings

We used 2016 county-level prevalence rates of diagnosed HIV infection rates for Latinos and non-Latino Whites obtained from the National HIV Surveillance System and factors obtained from multiple publicly available datasets. We used mixed effects Poisson modeling of observed HIV prevalence at the county-level to identify county-level factors that explained homogeneous effects across race/ethnicity and differential effects for Latinos and NL-Whites. Overall, the median Latinos disparity in HIV prevalence is 2.4; 94% of the counties have higher rates for Latinos than non-Latinos, and one-quarter of the counties' disparities exceeded 10. Of the 41 county-level factors examined, 24 showed significant effect modification when examined individually. In multi-variable modeling, 11 county-level factors were found that significantly affected disparities. Factors that increased disparity with higher, compared to lower values included proportion of HIV diagnoses due to injection drug use, percent Latino living in poverty, percent not English proficient, and percent Puerto Rican. Latino disparities increased with decreasing percent severe housing, drug overdose mortality rate, percent rural, female prevalence rate, social association rate, percent change in Latino population, and Latino to NL-White proportion of the population. These factors while significant had minimal effects on diminishing disparity, but did substantially reduce the variance in disparity rates.

### Conclusions

Large differences in HIV prevalence rates persist across almost all counties even after controlling for county-level factors. Counties that are more rural, have fewer Latinos, or have

documentation/national-data-documentation-2010-2018; NCHHSTP AtlasPlus available at: https://www.cdc.gov/nchhstp/atlas/index.htm; American Fact Finder available at: https://factfinder.census.gov/faces/tableservices/jsf/pages/productview.xhtml?pid=ACS_17_5YR_S1701; https://factfinder.census.gov/faces/tableservices/jsf/pages/productview.xhtml?pid=ACS_15_SPT_B01003; http://www.census.gov/popest/; https://factfinder.census.gov/faces/tableservices/jsf/pages/productview.xhtml?pid=PEP_CC-EST2016-ALLDATA2&prodType=document; https://www.census.gov/programs-surveys/acs/technical-documentation/table-and-geography-changes/2017/5-year.html.

**Funding:** This study was supported through funding from the: National Institute on Drug Abuse (NIDA, P30DA027828, NDB, CHB, DAA); https://www.drugabuse.gov/ National Institute of Allergy and Infectious Diseases (NIAID, P30AI7943, NDB); https://www.niaid.nih.gov/. The funders had no role in study design, data collection and analysis, decision to publish, or preparation of the manuscript. The content of this article is solely the responsibility of the authors and does not necessarily represent the official views of the funding agencies.

**Competing interests:** The authors have declared that no competing interests exist.

lower NL-White prevalence rates tend to have higher disparities. There is also higher disparity when community risk is low.

## Introduction

Latinos/Hispanics in the U.S. are disproportionately affected by HIV. Although Latinos comprise 18% of the population, they accounted for 25% of newly diagnosed HIV infections in 2016 [1]. The Latino rate of newly diagnosed HIV infection in 2016 was 17.0 per 100,000, which was three times greater than the 5.1 rate for non-Latino-Whites (NL-Whites). Similarly, the prevalence rate of diagnosed HIV infection for Latinos in 2016 was 372.1 per 100,000, a rate nearly 2.5 times that of NL-Whites (152.8 per 100,000).

Achieving viral suppression has important prevention and health benefits. People living with HIV with undetectable viral load can live healthy lives with somewhat higher rates than the general population of common and treatable age-related immune conditions [2], and are significantly less likely of transmitting HIV to others [3]. To reach viral suppression, people living with HIV need to follow steps along the HIV care continuum, starting with awareness of their HIV status, linkage and engagement in on-going HIV care, and adherence to HIV medication [4]. Latinos generally have poorer outcomes along the HIV care continuum than NL-Whites. According to the most recent report from the Centers for Disease Control and Prevention (CDC), the percentage of people living with HIV who are unaware of their HIV status is higher among Latinos compared with NL-Whites (16.7% vs. 11.5%) [5]. Latinos are nearly equal to NL-Whites in the percentage linked to HIV medical care within one month of HIV diagnosis (79.3% vs. 81.3%), while the percentage who have received any medical care (71.9% vs. 77.8%%), and the percentage living with diagnosed HIV infection who are virally suppressed (61.3% vs. 67.8%) are lower for Latinos. Latinos' delayed entry into care resulting from undiagnosed HIV infection, and lower percentage of viral suppression may partially explain Latino HIV incidence and prevalence disparities. However, community-level factors can also be associated with Latino disparities. This paper examines the role of multiple social and demographic factors that may help explain variations in Latino HIV prevalence and disparities in the US at the county-level.

Community-level factors can have a significant impact on differential access and engagement in HIV prevention and care services. County-level analyses offer the opportunity of rich socio-demographic and health data, and are often the seat of local government and resources. A number of studies have used area-based analyses of HIV surveillance data to identify factors associated with poor HIV outcomes and disparities [6–11], and some have focused on racial/ethnic HIV incidence and prevalence disparities that provide detailed results for Latinos [7, 12, 13]. Vaughn and colleagues conducted a county-level analysis of racial/ethnic HIV disparities in 2009 prevalence rates of diagnosed HIV infection across levels of urbanization, with a specific focus on how poverty interacted with urbanicity. In their analysis of 643 counties, Latinos had twice the rate of people living with diagnosed HIV infection compared to NL-Whites in large central metro counties, but these disparities disappeared in counties that had high levels of poverty. Interestingly, Latino disparities persisted after adjusting for poverty in mid-size and small counties. In another study [14], An and colleagues examined the association between HIV disparities in diagnosis rates and county-level socioeconomic position (SEP), a measure representing education, income and employment. Overall, Latinos had nearly twice the HIV diagnosis rate as NL-Whites (NL-White), and disparities became significantly more

pronounced with decreasing levels of SEP. However, similar to Vaughn's results, disparities persisted after controlling for SEP and urbanicity.

While county-level poverty and socio-economic position are associated with Latino/NL-White disparities, other moderating factors need to be considered to better explain the possible reasons for the observed disparity. In this paper we examine which, among a broad array of county-level social, economic, demographic, epidemiologic and access to care factors, affect Latino/NL-White HIV prevalence disparities, and seek to characterize the extent to which variations in HIV prevalence by race/ethnicity can be explained based on a combination of county-level factors.

## Methods

### Outcome variable

Prevalence rates of diagnosed HIV infection rates for Latinos (of any race) and NL-Whites (referred to as NL-White hereafter), were obtained from AIDSVu.org, which provides the latest publicly-available county-level data [15]. Data from AIDSVu.org originate from the CDC's National HIV Surveillance System (NHSS) [16] and reflect the number of persons aged 13 years or older who were living with diagnosed HIV infection through the end of 2016, which are based on CDC reports through June 2018 and adjusted for missing risk-factor information. County determination is based on residence at time of earliest HIV diagnosis. AIDSVu prevalence rate denominators were obtained by CDC from the U.S. Census Bureau.

To protect data confidentiality, CDC suppressed county-level race/ethnicity specific data with small numbers of people with diagnosed HIV infection (numerator) or, less frequently, small population estimates (denominator). Thus, counties included in our analyses met the following criteria: the number of race/ethnicity- specific prevalence of diagnosed HIV infection was 5 or more; the race/ethnicity-specific population estimate was at least 1000 persons aged 13 years or older; and the CDC had authorization from the state to release estimated prevalence counts for individual race/ethnicity groups.

Of the 3,142 counties in the US in 2016, 775 (25%) met the above criteria. These counties included 192,045 Latinos and 234,013 NL-White living with diagnosed HIV infection. Latinos and NL-Whites in these counties accounted for 90% and 77% of all people living with HIV infection the US and 88% and 66% of the US population, respectively.

### County level factors

We included publicly-available, county-level variables that had a potential association with HIV morbidity and that could serve as potential explanatory factors affecting disparity in prevalence of diagnosed HIV infection between Latinos and NL-White. Potential factors were identified from seven domains: (1) HIV characteristics, (2) Socioeconomic, (3) Community environment, (4) Health behaviors, (5) Access to health care, (6) Latino characteristics, and (7) Latino/non-Latino White ratios of county-level data such as the proportional difference in their respective populations in the county. For example, counties with smaller proportions of Latinos may experience different access to care compared to those with higher proportions of Latinos.

S1 Table describes 41 county-level factors, categorized into these seven domains, and provides the sources [15, 17–22] from which they were obtained. Wherever possible, data for each county were obtained for 2016 or the year closest to 2016 for which data were available.

## Statistical analysis

In this section we provide an overview of analytic methods, leaving details for later in the Methods section. We conducted exploratory analyses to assess the extent of Latino disparities, and then analyzed each county-level factor individually to determine whether it explained no variation in HIV diagnosis rates, explained similar effects across ethnicity, or showed different effects for Latinos and NL-White (e.g. moderation or effect modification). Factors with right-skewed distributions (e.g., prevalence rates, median household income, etc.) were log transformed after checking for nonlinearity with (log) prevalence, and this transformation was noted in tables. We confirmed that these transformations improved linearity by using non-parametric smoothers with quasi-Poisson regression models. We examined distributions and predictors of the prevalence rates for each race/ethnicity and used the Prevalence Risk Ratio (PRR), or the ratio of the Latino HIV prevalence to NL-White HIV prevalence in both exploratory and more formal models. We report PRRs as rate comparisons for Latinos versus NL-White with values larger than one indicating higher prevalence rates for Latinos than for NL-White.

By including multiple factors in the same analysis, we then examined unique contributions of the factors to prevalence and disparities. Comparison across these models allowed us to distinguish how much of a shared relationship exists between Latinos and NL-White around each factor as well as by a different or moderating relationship with these variables. We classified each factor's role in HIV prevalence in three ways. First, a county-level factor could have no effect on either overall prevalence or on disparities. Second the level of a factor could affect HIV prevalence equally for Latinos and NL-Whites, which we refer to as having a homogeneous effect of race/ethnicity on prevalence disparity (see S2 Technical Appendix sections III and IV in S1 Text). Third, county-level factors, could affect Latinos' and NL-Whites' HIV prevalence rates differentially; we refer to as factors having an effect modification or moderating effect, as the factor may interact with race/ethnicity to change disparities across levels of the factor. Finally, we examine the extent to which county-level variables explain or leave unexplained Latino HIV prevalence disparities.

## Exploratory analyses

We began by examining both the overall variation in Latino disparities across the counties, as well as the relationship Latino disparities had with NL-White prevalence. We used exploratory tools including plots and nonparametric smoothers to transform variables and to help guide our final model-building regarding how relationships with both Latino prevalence and disparity depend on combinations of the 41 county-level factors.

We examined the univariate and bivariate distribution of observed prevalence of diagnosed HIV infection rates for Latinos and NL-White. To examine the correlation between the two prevalence estimates on a logarithmic scale, we adjusted for sampling or measurement error using the Delta method and estimated the correlation between the "true scores," (i.e., accounting for sampling error; see S2 Technical Appendix section I in S1 Text). To examine how county-level factors are associated with Latino prevalence of diagnosed HIV infection and Latino HIV disparities relative to NL-Whites, we investigated whether non-linear relationships between each predictor and the observed Latino/NL-White log PRR exist, using locally weighted scatterplot smoothing (loess) [23]. If predictors were nonlinearly related on the log PRR scale, we transformed the factor to form a linear relationship; in all cases a logarithm was suitable.

## Single variable modeling of Latino HIV prevalence and Latino disparities in prevalence rates of diagnosed HIV infection

We used generalized linear mixed models (GLMMs, see details in the S2 Technical Appendix in S1 Text) [24] for analyses of county-level factors in single variable as well as in multi-variable analyses.

Prevalence rates of diagnosed HIV infection were fit using mixed effect Poisson modeling of observed counts (on a logarithmic scale) simultaneously for both ethnic groups at the county-level, with the log of each ethnic-specific population as an offset term. We described these as bivariate modeling since Latino and NL-White rates are analyzed in the same model and allowed to correlate with one another. Counties are treated as independent in our models, and we include random effects for each racial/ethnic group, within counties, to allow for extra-Poisson variation. We allow the two county-level random effects, one for Latinos and one for NL-White, to be correlated with one another so that we could model each racial/ethnic group's prevalence rate of diagnosed HIV infection separately as a function of single or multiple factors, as well as use this same model to account for log PRR disparity itself, which is the difference in the two log rates (see S2 Technical Appendix Part II in S1 Text).

Our models can be expressed as follows:

$$\log(Y_{ij}) = \beta_0 + \beta_1 NL_{ij} + \beta_2 X_i + \beta_3 NL_{ij}X_i + b_{0i} + b_{1i}NL_{ij} + \log(pop_{ij}) + \varepsilon_{ij} \qquad (1)$$

where, for convenience sake, NL is coded 0 for Latino and -1 for NL-White, and $X_i$ *is a factor*, $\beta s$ are fixed effects (described below), measuring homogeneity and disparity, and $b_{0i}$ and $b_{1i}$ are county-specific random effects for each ethnicity that follow a bivariate normal distribution with mean 0 and variance-covariance matrix $\Sigma$. These random effects allow for extra Poisson variation across counties, i.e., variation not explained by the covariates and sampling variation. The correlation between these two coefficients allows for the possibility that an individual county's Latino and NL White prevalence rates of diagnosed HIV infection could both be higher or lower than that predicted by the model. The specific coding allows for the following interpretation. If $\beta_3 = 0$, the coefficient $\beta_2$ measures the change in the log prevalence rate for both Latinos and NL Whites when $X_i$ changes one unit. The coefficient $\beta_3$ measures the change in the rate for Latino versus NL Whites (i.e. the disparity) when $X_i$ changes one unit. If $\beta_3$ differs from 0, $\beta_2$ measures the change in the log prevalence rate for Latinos.

To examine the homogeneous effect of a factor on both Latino and NL-White prevalence of diagnosed HIV infection, we include this factor term as having the same main effect across these two racial/ethnic groups. Because of the way we coded the data (see S2 Technical Appendix in S1 Text), a significant effect on the NL log rate coefficient $\beta_2$ and a non-significant interaction term is equivalent to a model having a homogeneous effect. For a moderating effect of how the factor's effect varies by Latino and NL-White prevalence of diagnosed HIV infection, the corresponding interaction parameter $\beta_3$ needs to be significant regardless of whether $\beta_2$ is (see S2 Technical Appendix Part III in S1 Text).

We used GLMMs to examine both the common (i.e., affecting Latinos and non-Latinos equally) and race/ethnicity-specific effects of factors on prevalence rates of diagnosed HIV infection. Previous publications and our own exploratory findings point to variation in the relations between both urbanicity and region with Latino disparity [12, 25]; thus both urbanicity and region are included as controlling variables in all GLMMs, along with their corresponding interaction terms with ethnicity. We also controlled for an AIDSVu county-level indicator called 'correctional warning' [26]. As individuals were assigned county based on current residence, and some rural counties had significant numbers of incarcerated Latinos in prisons, such counties could have an effect on our disparity rates.

We began by investigating the individual effects of 41 county-level factors. Our single factor GLMMs in Eq (1) above (e.g., ones that include say, percent unemployed and its interaction with a binary (0/1) ethnicity index) used data from all 775 counties if there was no suppression or missing data on that factor. With both a main effect and interaction term, this model examined whether the factor was associated with Latino HIV prevalence as well as with disparities. We used Wald-type tests and standardized regression coefficients to identify predictors with the largest effect on Latino prevalence of diagnosed HIV infection and Latino-NL-White disparity. To account for multiple comparisons among the single factor models, we identified predictors significantly associated with Latino prevalence of diagnosed HIV infection and Latino-NL-White disparity using a false discovery rate (FDR) of 5% [27].

## Multiple GLMM fitting of Latino HIV prevalence and Latino disparities in prevalence rates of diagnosed HIV infection

We then fit a multiple regression bivariate (Latino and NL-White) GLMM that includes all predictors found to be significantly associated with the Latino-NL-White disparity in the single variable models. Predictors that were collinear or highly correlated with other combinations of variables were excluded from the multi-variable bivariate GLMM.

A series of increasingly complex multiple bivariate GLMMs were fit in order to distinguish the effects of county-level main effects and moderators as well as two county-level Latino disparity indices—specifically the ratio of the county's population that is Latino versus NL-White and the ratio of the Latino versus NL-White populations that were impoverished. Including the last two factors and their interactions allows us to examine whether Latino versus NL-White demographic differences in population and poverty affected HIV disparities. Nested and non-nested models were compared using Bayesian Information Criterion (BIC) [28]. The proportion of unexplained deviance among Latino-NL-White disparity was computed for each multiple bivariate GLMM and for hierarchically nested models the percentage of the unexplained deviance that was explained by a more complex model is presented. Models with smaller unexplained deviance better explained disparity.

Each multi-variate model decomposes each county's (log) disparities into three parts, $\log(PRR_i) = \alpha_1 + \boldsymbol{\alpha_2}\boldsymbol{X_i} + \varepsilon_i$, corresponding to an overall difference in Latino versus NL-White log rates, $\alpha_1$, a portion of disparities explained by county-level factors, $\boldsymbol{\alpha_2}\boldsymbol{X_i}$, and a residual error in county-level disparities not explained by the model $\varepsilon_i$ after accounting for sampling error (see S2 Technical Appendix Part III in S1 Text). We decompose a model's unexplained and explained squared deviations (from 0) in Latino (log) disparity rates into two parts separated by parentheses as follows (E refers to the expected value averaged over sample values).

$$\text{Total Deviation from Zero} = (\ \alpha_1{}^2 + \text{Var}\varepsilon_i) + \text{E}(\boldsymbol{\alpha_2}\boldsymbol{X_i})^2 \qquad (2)$$

We centered all factors to have zero means so that the coefficient $\alpha_1$ always has the same interpretation as the average difference in log PRR. Note also that the overall disparity difference $\alpha_1{}^2$ is included in the unexplained part of the deviation as this coefficient adjusts for all factors in the model and would be zero if there were 100% moderation by these factors (see S2 Technical Appendix Part IV in S1 Text). To quantify the improvement of a more complex model to a simpler model, we compute the proportion of the unexplained deviation from zero in the simpler model that is explained by the more complex model, i.e., that shift accounted for by adding additional explanatory terms in $\boldsymbol{\alpha_2}\boldsymbol{X_i}$.

To examine how sensitive our models were to how we treated suppressed data, we fitted additional multiple regression models by removing covariates with a large amount of suppressed data (e.g. percent of HIV diagnoses due to injecting drug use) thereby increasing the

number of counties in the analysis. As the reported prevalence rates were not available by county by age and ethnicity, we used national data to consider whether confounding by age could account for the observed Latino disparities (S2 Technical Appendix Section V in S1 Text).

We conducted regression smoothing and GLMMs using glmer in R statistical package lme4 (Version 3.5.3 https://cran.R-project.org) and measurement error corrections in Mplus (Version 8.2 https://www.statmodel.com/).

## Results

Unsuppressed data from a total of 775 counties were available to compare Latino and NL-White county-level prevalence rates of diagnosed HIV infection. The median county-level prevalence rates of diagnosed HIV infection for Latinos and NL-White were 284 and 112 per 100,000, respectively, or 2.5 higher median rate for Latinos compared to NL-Whites. The median percent of Latinos living in these counties was 9%, and among Latinos, the median percent who were Mexican was 66.7%; 5.7% were Puerto Rican, and 27.6% were of other Latino origin. The Midwest, which includes 34% of all US counties, accounts for only 17% of analysis counties. The Northeast, South, and West, which represent 7%, 45%, and 14% of all US counties, respectively, account for 15%, 50%, and 19% of analysis counties, respectively. Thirty-two percent of the 775 counties are large metro areas (large central and large fringe), 23% are medium metro, 19% are small metro, 19% are micropolitan, and 7% are non-core.

Fig 1 plots the empirical prevalence rates of Latinos versus that for NL-White on logarithmic scales. The Y = X line indicates a PRR of 1, or no disparity, while points above this line represent counties with higher rates of HIV experienced by Latinos compared to NL-White, i.e. disparities. In the vast majority of counties, Latinos had higher prevalence rates of diagnosed HIV infection than did NL-White; 94% (n = 726) of counties had a Latino-NL-White prevalence rate ratio of diagnosed HIV infection (PRR) > 1. The mean and median Latino-NL-White PRR were 4.1 and 2.4, respectively. Only 6% (n = 49) of counties had observed PRR rates below 1. Furthermore, 33% (n = 257) had disparities between 1 and 2; 36% (n = 278) between 2 and 4, 19% (n = 145) between 4 and 10, leaving 6% (n = 46) of counties having a PRR > 10.

The correlation between prevalence rates for Latinos versus NL-White was 0.39 after correcting for Poisson sampling (measurement) error. The dashed line on Fig 1 is a nonparametric loess smoother showing how the Latino rate can be predicted from that of the NL-White rate. Because the curve deviates most from the Y = X line when the NL-White prevalence is low, the disparity tends to be higher when the NL-White rate is low and is negligible when the NL-White rate is extremely high. By examining where the loess smoother intersects with diagonal lines, we find that Latino disparities generally decrease with increased NL-White prevalence: the non-linear smoother shows that at the highest NL-White prevalence, Latino disparity routinely reaches 1, while among counties with low NL-White prevalence the Latino disparity routinely reaches 2 or greater.

### Region, urbanicity, and counties with significant correctional populations

Fig 2 illustrates the differences in median PRR across regions. The Northeast counties have the highest median PRR = 6.6, as well as the largest PRRs. The Midwest region has a median PRR of 3.2, the South median PRR is 2.1, and the West median PRR is 1.3. Twenty-one counties in the South, 27 in the West and 1 county in the Midwest have a PRR<1, while no counties in the Northeast have a PRR<1. Twenty-three percent of counties (n = 181) have a correctional warning. The median and mean PRR for counties with a correctional warning are 2.7 and 6.6, respectively. Counties with a correctional warning are primarily in the South (61%), followed

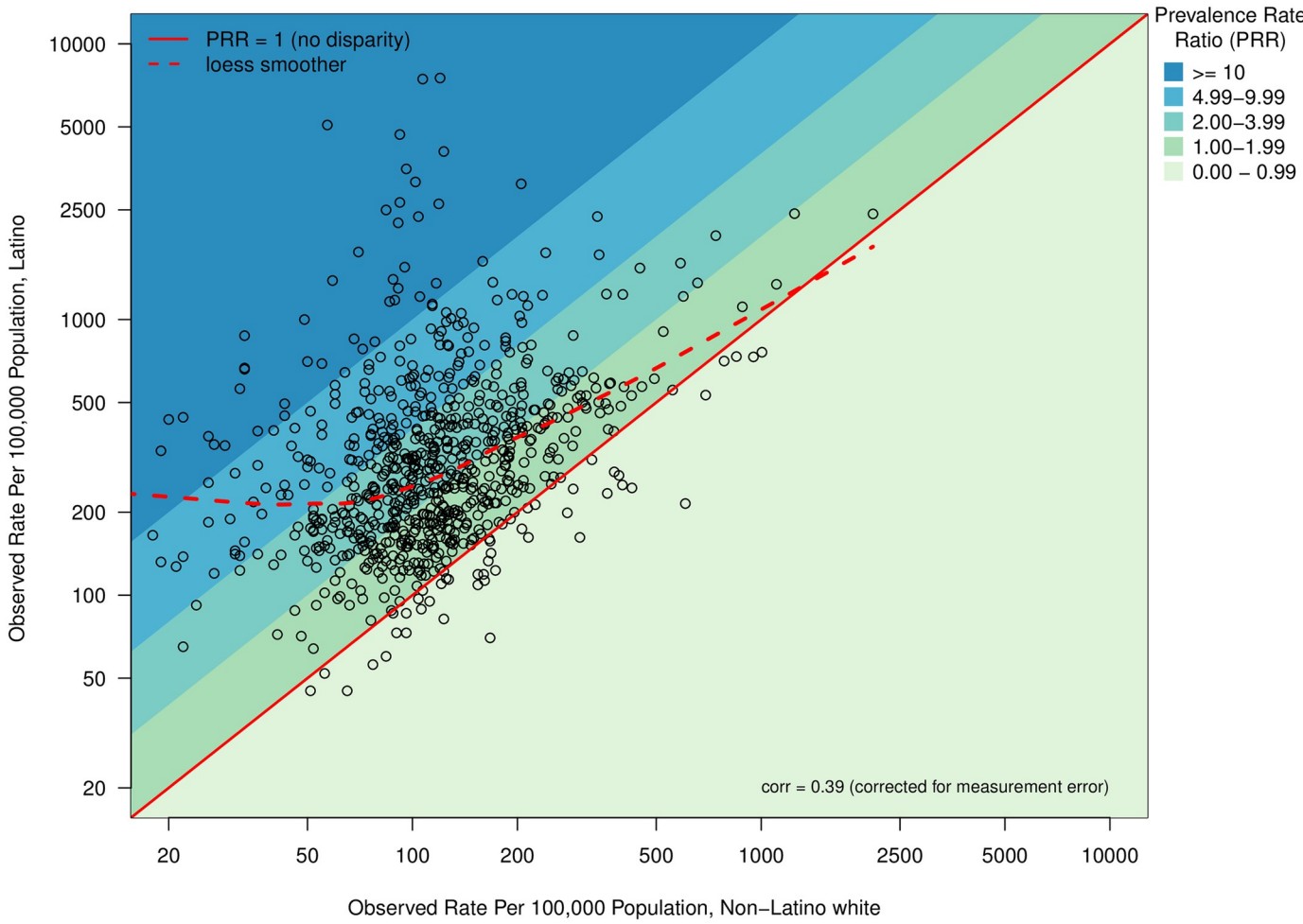

**Fig 1. Observed Latino and Non–Latino white diagnosed HIV prevalence rate (N = 775 counties).**

by the Northeast (19%), West (12%) and Midwest (6%). Of the twenty-two counties that have a PRR > 20, 16 have a correctional warning and nearly all are small counties in New York and Pennsylvania.

Fig 3 illustrates the change in median PRR across levels of urbanicity, and how this also varies by different levels of percent in poverty. Generally, disparities varied inversely with urbanicity, regardless of the total poverty percentage of the county. Counties with total percent in poverty less than 15% generally had the largest PRR, across all levels of urbanicity, compared to counties with higher percentages of poverty. As levels of percent in poverty increase, HIV disparities between Latinos and NL-Whites appear to remain stable; this was borne out by a non-significant 3-way interaction between urbanicity, poverty, and race/ethnicity.

## Single factor relationships

We evaluated single factor relationships including the main effect of the single factor plus its interaction with ethnicity after adjusting for main effects and interactions with region and urbanicity as well as presence of correctional institutions. Table 1 displays results from the 41 individual factor models both for Latino HIV prevalence and disparities. There were 11 factors

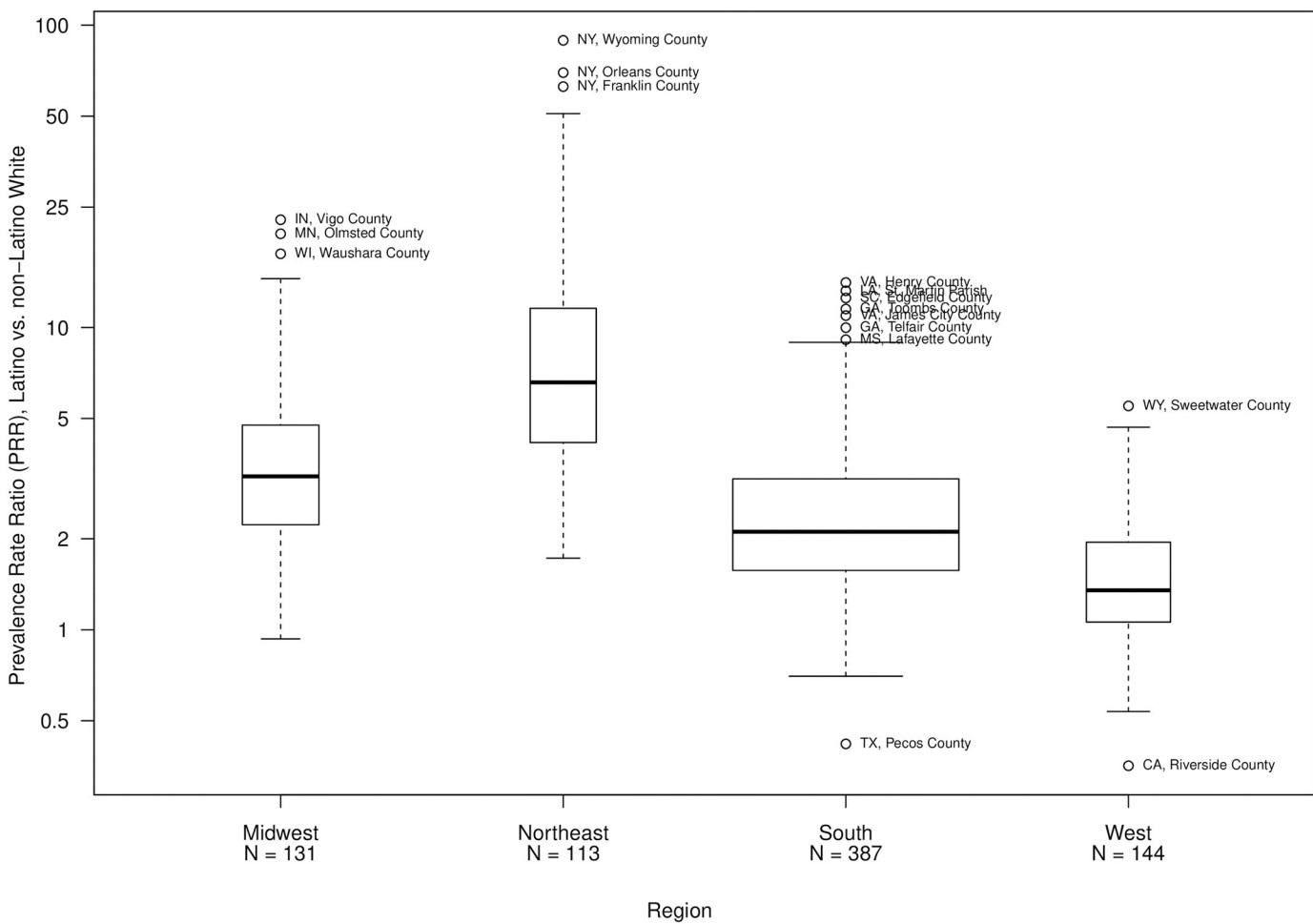

**Fig 2. Distribution of Latino and Non−Latino white diagnosed HIV prevalence rate ratio, by region (N = 775 counties).**

that showed homogeneous effects across ethnicity, 24 that showed effect modification by ethnicity, and 6 that were unrelated to either the prevalence of diagnosed HIV infection or disparity (accounting for 41 multiple tests with a false discovery rate of 5%).

### Factors individually related to Latino prevalence rate of diagnosed HIV infections

Of the 11 homogeneous factors, those with the largest positive effect on Latino log prevalence rate, based on standardized regression coefficients, were county prevalence rate ($\beta = 0.54$), male prevalence rate ($\beta = 0.52$), and injection drug use (IDU) prevalence rate ($\beta = 0.45$). Two homogenous factors had a negative effect: median household income ($\beta = -0.13$), and poverty ratio ($\beta = -0.06$).

### Factors related to Latino HIV prevalence and Latino disparities in prevalence rates of diagnosed HIV infection from single factor models

Of the 24 factors significantly associated with Latino-NL-White HIV disparities, those with the strongest positive effects were percent associated with injection drug use (IDU) ($\beta = 0.17$),

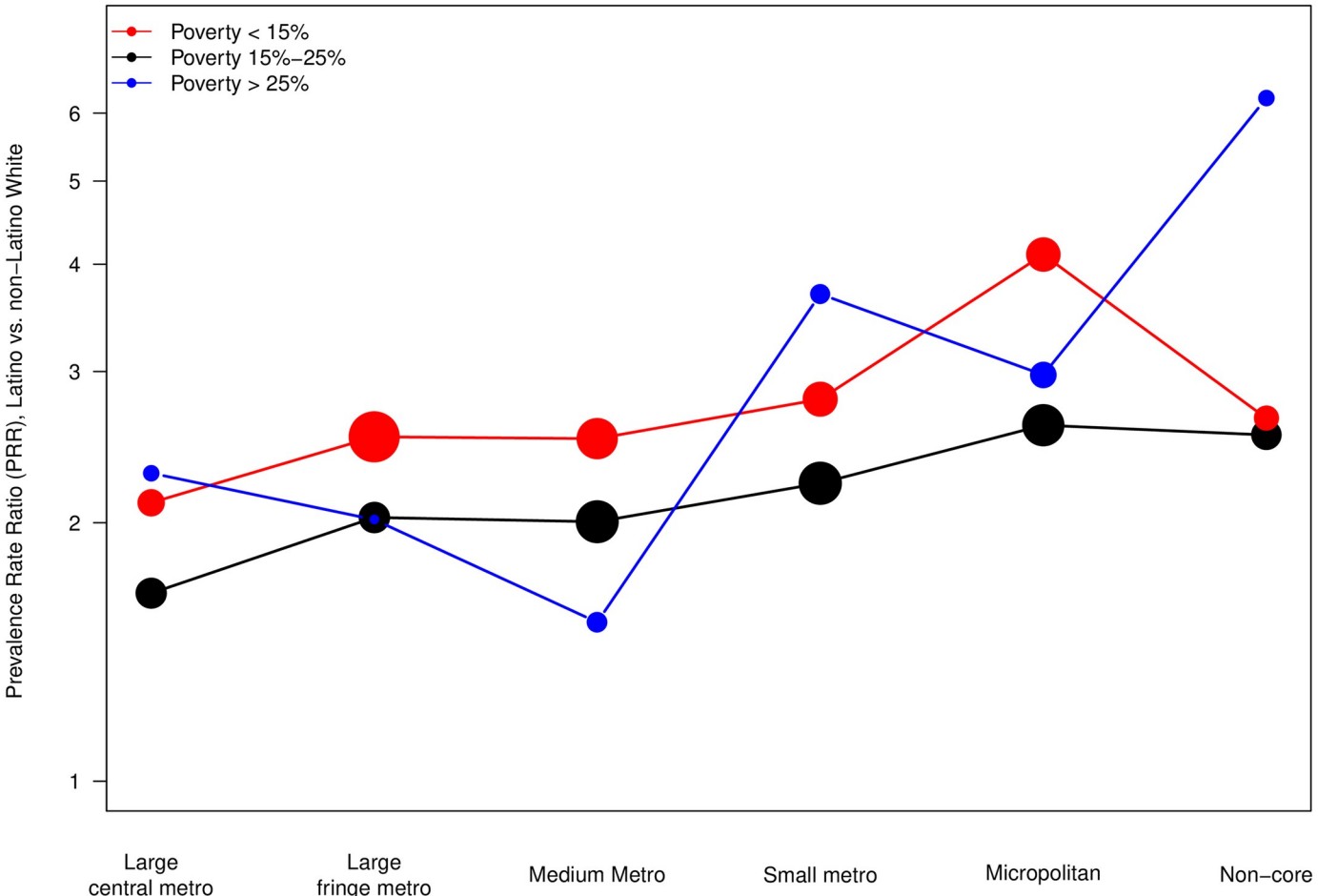

**Fig 3. Median Latino and Non–Latino white diagnosed HIV prevalence rate ratio by urbanicity within levels of percent in poverty within county (N = 775 counties).** The size of circle is proportional to number of counties within urbanicity and poverty level.

percent rural ($\beta = 0.16$), and percent with a high school education ($\beta = 0.14$), while population ratio ($\beta = -0.30$), percent uninsured ($\beta = -0.28$), and percent Latino ($\beta = -0.23$), had the largest negative effect.

There were six factors that that were unrelated to either the Latino prevalence of diagnosed HIV infection or disparity: percent heterosexual, percent unemployed, Black/White Segregation Index, percent excessive drinking, primary care provider rate, and income ratio.

## Multi-variable modeling of disparities

Table 2 examines two multi-variable models. Both models are adjusted for region, urbanicity, and their interactions with race/ethnicity; and are also adjusted for a county-level indicator on whether it had a correctional warning. Model 1 included all factors significantly associated with Latino disparities in the single factor models, excluding the following redundant/collinear variables: percent MSM, MSM prevalence rate while retaining percent IDU; and percent Latino, while retaining % Mexican and % Puerto Rican. Model 1 is estimated on 485 counties with complete data on the 21 factors included. Model 2 additionally excludes two HIV characteristics with large amounts of missing/suppressed data: percent IDU, and new HIV diagnosis

**Table 1. Individual covariate impact on Latino prevalence and disparities (maximum N = 775 counties).**

| Category | Variable | N | Latino HIV Prevalence | | | Disparity | | | Role of Factor |
|---|---|---|---|---|---|---|---|---|---|
| | | | Beta | Std[†] Beta | p-value | Beta | Std[†] Beta | p-value | |
| **HIV Characteristics** | County Prevalence Rate[β] | 775 | 0.74 | 0.54 | < 0.001* | 0 | 0.01 | 0.98 | Homogeneous |
| | Male Prevalence Rate[β] | 730 | 0.74 | 0.52 | < 0.001* | -0.04 | -0.02 | 0.3 | Homogeneous |
| | Female Prevalence Rate[β] | 730 | 0.28 | 0.27 | < 0.001* | -0.09 | -0.08 | < 0.001* | Effect Modifier |
| | % MSM | 730 | -0.02 | -0.18 | < 0.001* | -0.01 | -0.17 | < 0.001* | Effect Modifier |
| | % IDU | 656 | 0.02 | 0.19 | < 0.001* | 0.02 | 0.17 | < 0.001* | Effect Modifier |
| | % Heterosexual | 744 | 0 | -0.02 | 0.35 | 0 | 0.03 | 0.31 | |
| | New Diagnosis Rate[β] | 549 | 0.11 | 0.16 | < 0.001* | -0.05 | -0.08 | 0.01* | Effect Modifier |
| | IDU Prevalence Rate[β] | 656 | 0.47 | 0.45 | < 0.001* | 0.05 | 0.05 | 0.05 | Homogeneous |
| | MSM Prevalence Rate[β] | 774 | 0.73 | 0.51 | < 0.001* | -0.08 | -0.05 | 0.02* | Effect Modifier |
| **Socioeconomic** | Median Household Income[β] | 775 | -0.52 | -0.13 | < 0.001* | 0.03 | 0.01 | 0.79 | Homogeneous |
| | GINI Index | 775 | 0.04 | 0.14 | < 0.001* | -0.02 | -0.06 | 0.00* | Effect Modifier |
| | % High School Education | 775 | 0.01 | 0.04 | 0.1 | 0.02 | 0.14 | < 0.001* | Effect Modifier |
| | % Unemployed | 775 | 0.01 | 0.02 | 0.49 | -0.02 | -0.03 | 0.19 | |
| | Income Inequality Ratio[β] | 775 | 0.97 | 0.14 | < 0.001* | -0.35 | -0.05 | 0.02* | Effect Modifier |
| | Non-White/White Segregation Index | 775 | 0.01 | 0.1 | < 0.001* | 0 | 0.05 | 0.04 | Homogeneous |
| | Black/White Segregation Index | 754 | 0 | 0.02 | 0.5 | 0 | -0.01 | 0.74 | |
| | % Non-Latino White Living Below Poverty | 75 | 0.03 | 0.14 | < 0.001* | 0.01 | 0.05 | 0.05 | Homogeneous |
| | % Living Below Poverty | 775 | 0.02 | 0.14 | < 0.001* | 0 | -0.02 | 0.33 | Homogeneous |
| **Community Environment** | % Single Parent Household | 775 | 0.02 | 0.2 | < 0.001* | 0 | -0.01 | 0.52 | Homogeneous |
| | Social Association Rate | 775 | 0.13 | 0.05 | 0.07 | 0.24 | 0.08 | < 0.001* | Effect Modifier |
| | Violent Crime Rate | 767 | 0.09 | 0.06 | 0.02* | -0.25 | -0.16 | < 0.001* | Effect Modifier |
| | % Severe Housing Problems | 775 | 0.02 | 0.09 | < 0.001* | -0.03 | -0.15 | < 0.001* | Effect Modifier |
| | % Rural | 775 | 0 | 0.03 | 0.4 | 0.01 | 0.16 | < 0.001* | Effect Modifier |
| **Health Behaviors** | % Excessive Drinking | 775 | -0.01 | -0.04 | 0.08 | 0 | -0.01 | 0.75 | |
| | Chlamydia Rate[β] | 775 | 0.21 | 0.12 | < 0.001* | -0.05 | -0.03 | 0.18 | Homogeneous |
| | Drug Overdose Mortality Rate[β] | 706 | -0.02 | -0.01 | 0.73 | -0.12 | -0.05 | 0.01* | Effect Modifier |
| | Gonorrhea Rate[β] | 775 | 0.12 | 0.11 | < 0.001* | -0.08 | -0.07 | 0.00* | Effect Modifier |
| **Access to Health Care** | % Uninsured | 775 | -0.03 | -0.15 | < 0.001* | -0.06 | -0.28 | < 0.001* | Effect Modifier |
| | Primary Care Provider Rate[β] | 774 | 0.03 | 0.02 | 0.51 | -0.05 | -0.03 | 0.2 | |
| | Mental Health Provider Rate[β] | 771 | 0.14 | 0.12 | < 0.001* | 0 | 0 | 0.92 | Homogeneous |
| | Preventable Hospitalization Rate[β] | 774 | -0.1 | -0.03 | 0.25 | -0.24 | -0.07 | 0.00* | Effect Modifier |
| | Healthcare Costs[β] | 775 | -0.81 | -0.11 | < 0.001* | -0.95 | -0.13 | < 0.001* | Effect Modifier |

*(Continued)*

**Table 1.** (Continued)

| Category | Variable | N | Latino HIV Prevalence | | | Disparity | | | Role of Factor |
|---|---|---|---|---|---|---|---|---|---|
| | | | Beta | Std† Beta | p-value | Beta | Std† Beta | p-value | |
| **Latino Characteristics** | Latino Population % Change 2000–2014 | 774 | -0.05 | -0.05 | 0.04 | 0.1 | 0.1 | < 0.001* | Effect Modifier |
| | % Mexican | 775 | -0.01 | -0.21 | < 0.001* | 0 | -0.1 | < 0.001* | Effect Modifier |
| | % Puerto Rican | 775 | 0.01 | 0.17 | < 0.001* | 0.01 | 0.13 | < 0.001* | Effect Modifier |
| | % Latino | 775 | -0.01 | -0.14 | < 0.001* | -0.01 | -0.23 | < 0.001* | Effect Modifier |
| | % Not English Proficient | 775 | -0.02 | -0.05 | 0.02* | -0.05 | -0.17 | < 0.001* | Effect Modifier |
| | % Latino Living Below Poverty | 775 | 0.01 | 0.09 | < 0.001* | 0.01 | 0.09 | < 0.001* | Effect Modifier |
| **Latino/Non-Latino White Ratios** | Poverty Ratio (% Latino / % Non-Latino White) β | 775 | -0.13 | -0.06 | 0.01* | 0.03 | 0.01 | 0.59 | Homogeneous |
| | Population Ratio (% Latino / % Non-Latino White) β | 775 | -0.13 | -0.16 | < 0.001* | -0.26 | -0.3 | < 0.001* | Effect Modifier |
| | Income Ratio (% Latino / % Non-Latino White) β | 766 | -0.22 | -0.04 | 0.05 | -0.2 | -0.04 | 0.05 | |

†Std = standardized.

β Denotes log transformation of variable.

* Denotes significance using a False Discovery Rate (FDR) of 5%.

rate. A total of 675 counties with complete data on the remaining 19 factors were included in Model 2.

There are few differences in estimates and significance between the two models. Eleven factors were significantly associated with disparities in Model 1, eight of which were also significant in Model 2. Percent high school education is negatively associated with Latino disparities in Model 2 but it is not significantly associated in Model 1. Female prevalence rate and percent rural has a significant negative association in Model 1 but is not significant in Model 2. In both multi-variable models, the % Latino / % NL-White population ratio has the largest negative effect on Latino disparities. In Model 1, percent IDU has the largest positive effect on Latino disparities ($\beta = 0.021$), after adjusting for all other factors. Percent Puerto Rican has the largest positive effect in Model 2. Results are similar in both models, but the overall disparity was smaller in model 1 (adjusted PRR = 2.40) compared to model 2 (adjusted PRR = 2.68). The 290 counties excluded in Model 1 have higher prevalence (median PRR = 2.9) and are smaller counties with smaller Latino population.

Model 1 was selected for further analyses due to its additional variables describing county HIV characteristics which allows us to examine their added contributions.

All significant moderator effects in Model 1 are shown in Fig 4. As counties' percent of diagnoses associated with injection drug use increase, the disparities ratio nearly doubles, due to increasing prevalence for Latinos and no increase for NL-White. Similarly for percent not English proficient, the disparity nearly doubles, mostly due to a sharper increase in Latino prevalence than that of NL-White prevalence. An even larger interaction occurs with the population ratio of Latinos to NL-Whites where disparity increases as the proportion of Latinos decreases. Counties with a smaller proportion of Latinos have a Latino prevalence rate nearly four times that of NL-Whites compared to 1.5 times when there are higher proportions of Latinos in the county. Overall, major interactions correspond to changes in Latino prevalence

**Table 2. A multivariate analysis of county level interactions involving Latino disparity.**

| Category | Variable[δ] | Model 1: N = 485[a] | | | | | Model 2: N = 675[b] | | | |
|---|---|---|---|---|---|---|---|---|---|---|
| | | Beta | Std[†] Beta | SE | p-value | Role of Factor | Beta | Std[†] Beta | SE | p-value |
| **HIV Characteristics** | County Prevalence Rate[β] | — | — | — | — | | — | — | — | — |
| | Male Prevalence Rate[β] | — | — | — | — | | — | — | — | — |
| | Female Prevalence Rate[β] | -0.07 | -0.06 | 0.03 | 0.044* | Effect Modifier | -0.02 | -0.02 | 0.03 | 0.574 |
| | % MSM | — | — | — | — | | — | — | — | — |
| | % IDU | 0.03 | 0.21 | 0 | < 0.001* | Effect Modifier | — | — | — | — |
| | % Heterosexual | — | — | — | — | | — | — | — | — |
| | New Diagnosis Rate[β] | 0.03 | 0.03 | 0.03 | 0.312 | Homogeneous | — | — | — | — |
| | IDU Prevalence Rate[β] | — | — | — | — | | — | — | — | — |
| | MSM Prevalence Rate[β] | — | — | — | — | | — | — | — | — |
| **Socioeconomic** | Median Household Income[β] | — | — | — | — | | — | — | — | — |
| | GINI Index | -0.01 | -0.02 | 0.01 | 0.629 | | -0.02 | -0.06 | 0.01 | 0.115 |
| | % High School Education | -0.01 | -0.04 | 0.01 | 0.223 | | -0.02 | -0.08 | 0.01 | 0.031* |
| | % Unemployed | — | — | — | — | | — | — | — | — |
| | Income Inequality Ratio[β] | 0.5 | 0.07 | 0.28 | 0.078 | Homogeneous | 0.81 | 0.11 | 0.28 | 0.004* |
| | Non-White/White Segregation Index | — | — | — | — | | — | — | — | — |
| | Black/White Segregation Index | — | — | — | — | | — | — | — | — |
| | % Non-Latino White Living Below Poverty | — | — | — | — | | — | — | — | — |
| | % Living Below Poverty | — | — | — | — | | — | — | — | — |
| **Community Environment** | % Single Parent Household | — | — | — | — | | — | — | — | — |
| | Social Association Rate | -0.2 | -0.07 | 0.07 | 0.007* | Effect Modifier | -0.14 | -0.05 | 0.07 | 0.037* |
| | Violent Crime Rate | -0.07 | -0.04 | 0.05 | 0.161 | | -0.1 | -0.06 | 0.04 | 0.033* |
| | % Severe Housing Problems | -0.03 | -0.13 | 0.01 | < 0.001* | Effect Modifier | -0.03 | -0.12 | 0.01 | < 0.001* |
| | % Rural | 0 | -0.06 | 0 | 0.046* | Effect Modifier | 0 | -0.02 | 0 | 0.625 |
| **Health Behaviors** | % Excessive Drinking | — | — | — | — | | — | — | — | — |
| | Chlamydia Rate[β] | — | — | — | — | | — | — | — | — |
| | Drug Overdose Mortality Rate[β] | -0.26 | -0.12 | 0.05 | < 0.001* | Effect Modifier | -0.24 | -0.11 | 0.05 | < 0.001* |
| | Gonorrhea Rate[β] | 0 | 0 | 0.03 | 0.921 | | -0.03 | -0.02 | 0.03 | 0.389 |
| **Access to Health Care** | % Uninsured | 0 | 0 | 0.01 | 0.932 | | 0 | -0.01 | 0.01 | 0.85 |
| | Primary Care Provider Rate[β] | — | — | — | — | | — | — | — | — |
| | Mental Health Provider Rate[β] | — | — | — | — | | — | — | — | — |
| | Preventable Hospitalization Rate[β] | -0.13 | -0.04 | 0.11 | 0.235 | | -0.07 | -0.02 | 0.1 | 0.527 |
| | Healthcare Costs[β] | -0.32 | -0.04 | 0.23 | 0.162 | Homogeneous | -0.38 | -0.05 | 0.23 | 0.096 |
| **Latino Characteristics** | Latino Population % Change 2000–2014 | -0.05 | -0.05 | 0.02 | 0.035* | Effect Modifier | -0.07 | -0.07 | 0.02 | 0.001* |
| | % Mexican | 0 | -0.07 | 0 | 0.052 | Homogeneous | 0 | -0.04 | 0 | 0.277 |
| | % Puerto Rican | 0.01 | 0.09 | 0 | 0.006* | Effect Modifier | 0.01 | 0.12 | 0 | < 0.001* |
| | % Latino | — | — | — | — | | — | — | — | — |
| | % Not English Proficient | 0.04 | 0.13 | 0.01 | 0.003* | Effect Modifier | 0.03 | 0.1 | 0.01 | 0.015* |
| | % Latino Living Below Poverty | 0.02 | 0.12 | 0 | < 0.001* | Effect Modifier | 0.01 | 0.09 | 0 | < 0.001* |

*(Continued)*

**Table 2.** (Continued)

| Category | Variable[δ] | Model 1: N = 485[a] | | | | | Model 2: N = 675[b] | | | |
|---|---|---|---|---|---|---|---|---|---|---|
| | | Beta | Std[†] Beta | SE | p-value | Role of Factor | Beta | Std[†] Beta | SE | p-value |
| **Latino/Non-Latino White Ratios** | Poverty Ratio (% Latino / % Non-Latino White) [β] | — | — | — | — | | — | — | — | — |
| | Population Ratio (% Latino / % Non-Latino White) [β] | -0.35 | -0.4 | 0.04 | < 0.001* | Effect Modifier | -0.33 | -0.38 | 0.03 | < 0.001* |
| | Income Ratio (% Latino / % Non-Latino White) [β] | — | — | — | — | | — | — | — | — |

[δ] Each row represents a factor in a multivariable model, only the interaction terms from the models are displayed. A dashed line indicates this variable was non-significant in the individual level analyses of disparity and therefore not included in this multivariate analysis, or they were redundant and excluded from the final model.

[†]Std = standardized.

[a] Model 1 includes all variables found significant in the individual level analyses, excluding redundant variables (% MSM, MSM Prevalence Rate while retaining % IDU; % Latino while retaining % Mexican and % Puerto Rican).

[b] Model 2 additionally excludes two HIV characteristics with large amounts of missing/suppressed data: % IDU, and New Diagnosis Rate.

[β] Denotes log transformation of variable.

* Significant at p<0.05.

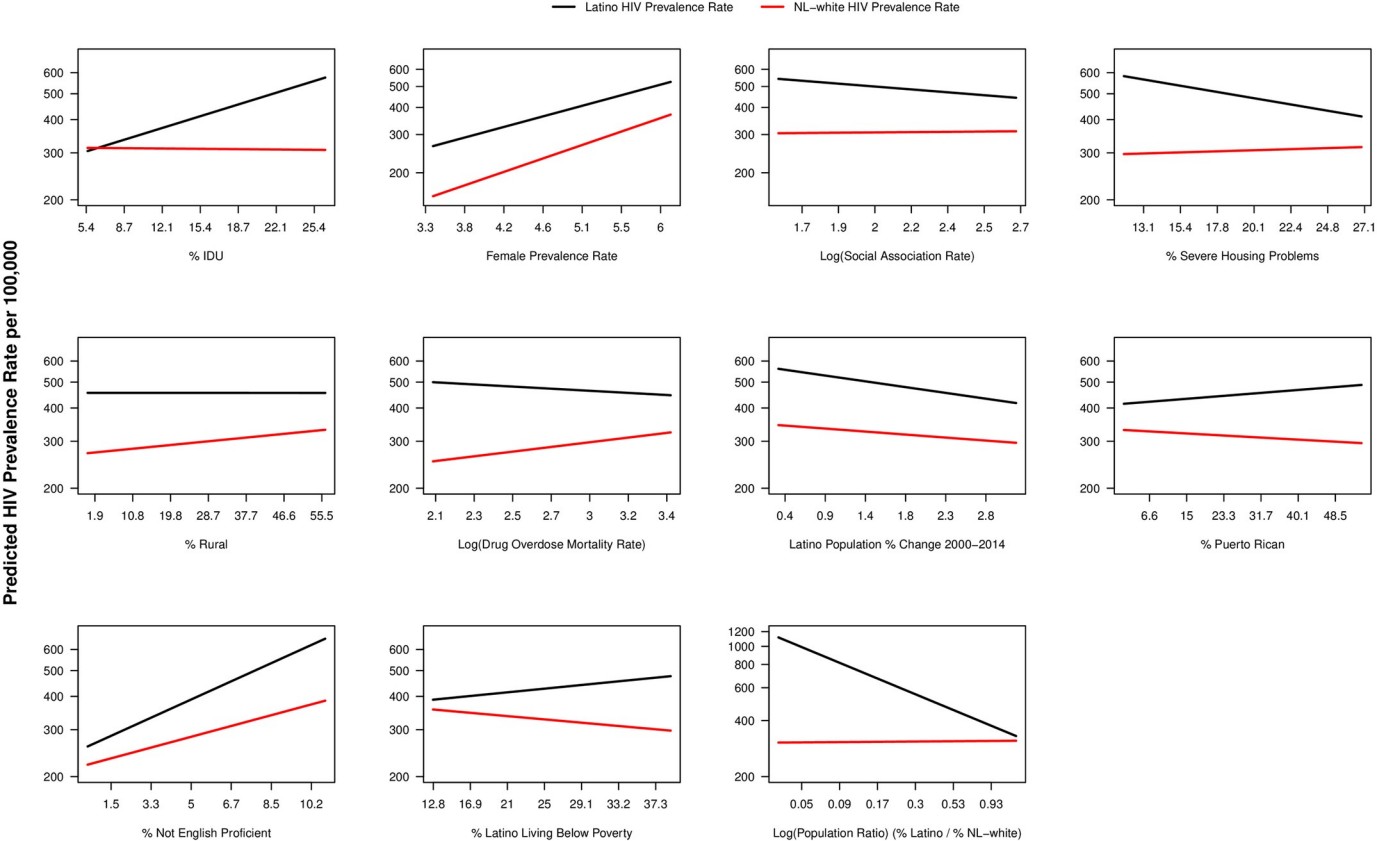

**Fig 4. Moderator effects on the estimated HIV prevalence rate for significant factors in multivariate model.**

rates with little to no change in NL-Whites, and disparities tend to be lower when overall rates for both Latinos and NL-Whites are extremely high.

## Disparity remaining unexplained with different factors

Table 3 compares a number of hierarchically nested multivariate regression models estimated using 485 counties and the factors in Model 1 above. It summarizes each model fit with degrees of freedom and BIC, which improves with model complexity. It also details each model's Latino disparity coefficient (log PRR) as well as each model's residual variance and total unexplained deviance in Latino disparity, based on which factors and interaction terms are included in the models. The bottom portion of the table compares models' unexplained deviance with values below the diagonal representing the proportion of unexplained deviance that is reduced by the more complex model. The first model (M0) includes no additional factors and shows that the average unadjusted PRR is 2.36, the mean (log PRR) is 0.86, the unexplained variance is 0.41, so the unexplained deviance is 1.15 (= $0.86^2$+ 0.41), or 64% percent accounted for by the mean. Model M1 includes region, urbanicity, and their interactions; this accounts for 21% of the unexplained deviance in Model 0. Inclusion of other factors from Table 2 Model 1 that have significant impact on prevalence and disparities contribute very little to explaining variation in disparities. Specifically, addition of homogeneous factors that significantly affect both race/ethnicity groups alike (M2) and race/ethnicity specific factors (M3) produce less than one percent reductions in the mean disparity or variance. Additional

**Table 3. Comparison of unexplained deviance in disparities in multi-variable models (N = 485 counties).**

| Model Fit | | M0: Unadjusted | M1: Adjusted for Region & Urbanicity Effect Modifiers | M2: M1 + Homogeneous (Non-Race/Ethnicity-Specific) Factors[1] | M3: M2 + Race/ Ethnicity-Specific Factors[2] | M4: M2 + Non-Race/Ethnicity-Specific Effect Modifiers[1] | M5: M3 + Race/ Ethnicity-Specific Effect Modifiers[2] |
|---|---|---|---|---|---|---|---|
| df Error | | 965 | 948 | 928 | 927 | 908 | 906 |
| BIC | | 10825 | 10385 | 10198 | 10195 | 10173 | 10092 |
| Mean[3] | | 0.86 | 0.86 | 0.86 | 0.86 | 0.86 | 0.87 |
| Adjusted PRR | | 2.36 | 2.36 | 2.35 | 2.36 | 2.37 | 2.4 |
| Residual Variance | | 0.41 | 0.17 | 0.18 | 0.18 | 0.11 | 0.09 |
| Total Unexplained Deviance[4] | | 1.15 | 0.91 | 0.91 | 0.91 | 0.86 | 0.85 |
| % Unexplained Deviance[5] | | 64% | 81% | 80% | 81% | 87% | 90% |
| Unexplained Deviance Across Models (Diagonals) and Percentage of Deviance Explained by Effect Modifiers (Off-Diagonal) | M0 | 1.15 | 21% | 21% | 20% | 25% | 26% |
| | M1 | | 0.91 | 0% | -1% | 5% | 6% |
| | M2 | | | 0.91 | 0% | 6% | 6% |
| | M3 | | | | 0.91 | 6% | 7% |
| | M4 | | | | | 0.86 | 1% |
| | M5 | | | | | | 0.85 |

[a]Models labeled with factors include just their main effects; models labeled with effect modifiers include main effects and interaction terms with ethnicity.

[1] Includes the following factors: Female prevalence rate, % IDU, New diagnosis rate, GINI index, % high school education, income inequality ratio, social association rate, violent crime rate, % severe housing problems, % rural, drug overdose mortality rate, Gonorrhea rate, % uninsured, preventable hospitalization rate, healthcare costs, Latino population % change, % Puerto Rican, % not English proficient; % Latino living below poverty.

[2] Includes the following factors: Population Ratio (% Latino/ % NL-White).

[3] "Mean" coefficient represents the difference between Latino and NL- White log (PRR) after adjusting for county-level factors.

[4] "Total Unexplained Deviance" is the sum of "Mean" and "Residual Variance".

[5] Percent unexplained deviance due to difference between Latino and NL- White log (PRR).

inclusion of non-race/ethnicity effect modifiers (M4) do make significant reduction in the unexplained variance from 0.18 to 0.11, but have no impact on the adjusted PRR disparity that is still not explained. Model M5, which includes the ratio of Latino to NL-White population, adds only an additional 1% of explanation (change from 25% to 26% reduction in unexplained deviance). Thus the significant effect moderators contribute minimally to explaining the overall level of disparity while making a major reduction in the variance of disparities.

## Discussion

In almost every analysis county Latinos had higher prevalence rates of diagnosed HIV infection than NL-Whites, with larger disparities observed in counties with lower NL-White HIV prevalence rates and in small counties with a low Latino population. The median disparity across all counties with available data was 2.4. The Northeast region had counties with the highest disparities and 30% of them had a disparity rate greater than 10Counties in the West showed the lowest overall rate of disparity, with a median PRR of 1.3. Together, region and urbanicity, along with their interaction, explained 21% of the deviance in disparities. Counties with low levels of percent living in poverty had higher rates of disparity than those with higher poverty rates when we controlled for level of urbanicity.

Our single variable analyses found more than half of the 41 county-level factors to be significantly related to Latino disparities, and most of these were related to Latino HIV prevalence as well. There were strong significant effects in these single variable analyses across all six categories. In multi-variate analyses, half of these factors remained significant, most of which (82%) were in categories that represented Latino demographic characteristics, community environment, and certain characteristics of HIV. For the two HIV characteristics, disparities increased in counties with higher percent of HIV diagnoses due to injecting drug use and decreased in counties with a higher female HIV prevalence rate. Four of the five included factors associated with Latino-specific characteristics remained significant. The percent not English proficient, percent Latino living below poverty, and percent Puerto Rican were positively related to disparities, and Latino population percent change was negatively related with disparities. Other county-level characteristics, including social association rate, percent severe housing, percent rural, drug overdose mortality rate, and population ratio (proportion of Latino to NL-White) were all inversely related with disparities. While region and urbanicity effect modification factors collectively explained 21% of disparities, most of the disparities remain unexplained. Almost all of this reduction in variation comes from explaining the variance in disparities, which decreased by 50% from 0.17 to 0.09.

We found that disparities for Latinos are pervasive, and that counties that are less urban, have fewer Latinos, or have lower NL-White prevalence rates tend to have higher disparity. After controlling for urbanicity and geographic regions, disparities are greater in counties with characteristics specific to Latinos, and become more pronounced with increases in the percent of Latinos below poverty, percent of Latinos who are Puerto Rican, and percent who are not English proficient. Conversely, counties with stronger community resources and cohesion, measured through proxies such as diminished housing problems and greater social association rates, and those with larger proportion of NL-White than Latino population, experience lower disparities.

Our findings are consistent with factors identified in the literature that are associated with Latino access to HIV prevention and care services that contribute to disparities, and suggest ways in which they may be addressed. Reduced English proficiency among Latinos has been associated with lower HIV testing rates [29, 30], challenges to establish ties with HIV providers [31], and finding providers who can speak their language [32]. Latinos in more rural areas

where HIV prevalence is typically low, have a higher percentage of late HIV diagnosis than those in more urban areas [33, 34]. In such areas, language barriers, coupled with potentially less HIV services, would differentially affect Latinos across both HIV prevention and treatment cascades [35]. There are a number of emerging interventions tailored for Latinos to address these barriers [36]. For instance, a culturally appropriate intervention composed of a bilingual and bicultural HIV care team found significant increases in the number of scheduled and kept visits 12-months after implementation [37]. Delivering HIV care service through Telemedicine could help fill the service gap in less urban areas. A recent review of the use of Telemedicine for PrEP in both rural and urban areas found high PrEP initiation and retention rates [38]. A bilingual/bicultural care team, adapted for delivery via Telemedicine to provide HIV care and PrEP services has the potential to increase access to care and reduce disparities.

Factors representing community environment and organization were significantly associated with disparities in our analysis, and are aligned with the socioecological model that identified community environment, organization, and social support as key factors that can facilitate or hinder Latino's ability to access and adhere to HIV care services [39, 40]. A study of Latino MSM in North Carolina, a state with a small but growing Latino population, found greater odds of HIV testing among those Latinos who had general and HIV-related social support [29]. Another qualitative study examining barriers to retention in care among Latino migrants and immigrants, and providers in Boston found that family support and trusting relationships with health care providers strongly influenced retention in care [31]. Both patients and providers valued the role of different team members, especially community health workers (CHW). A systematic review of US-studies that utilized CHWs to improve ART adherence found that peer education focused on medication management and daily observation of taking ART improved adherence [41]. Another systematic review of lay health worker interventions for Latinos conducted by Rhodes and colleagues identified 14 studies with positive evidence of effectiveness; however, only 2 had an HIV focus, neither of which specifically targeted men [42]. Further research is needed to identify the scope of services to be provided by community/lay health workers to address retention in HIV prevention and care services among Latinos, and as Thomas Painter suggests, explore ways to incorporate spontaneously occurring social support processes in Latino communities into the design of HIV prevention programs to build on community- based strengths (social assets) and complement individual-level interventions [43].

Intersectionality marginalizes Latinos who use injecting drugs as well as Latino men who have sex with men. These can translate into levels of stigma around racism, heterosexism, and mental health discrimination and bias not experienced by NL-White who use injecting drugs or are MSM [44–46]. Additionally, discrimination and acculturation have been found to be associated with substance use among young Latino [47, 48]. Such effects can be compounded when there is questioning of one's or family member's right to US residence, work, schooling, and access to health care during this time when America's views on immigration are politically volatile [49].

Our finding of higher HIV disparities with higher proportions of Puerto Ricans and proportion of cases due to injecting drugs at the county-level is in accord with a synthesis of data from multiple studies involving drug treatment-seeking adults [50] which found that both needle sharing and drug use severity were higher among Latinos than NL-Whites, increasing the likelihood of HIV transmission via injection drug use. National-level adolescent drug use rates have been consistently higher among Latinos than NL-White [51, 52], and in a review of Latino HIV risk, Loue [53] noted higher rates of HIV transmission through injecting drug use for Puerto Ricans both on the island as well as on the US mainland. As a potential solution, family-based prevention programs specifically designed for young Hispanics in Miami [54]

and Mexican Americans in Arizona [55], have demonstrated prevention of drug abuse and lower sexual risk behaviors.

Lastly, our finding of a strong association between disparities and percent of Latinos living below poverty, and interactions between disparities, urbanicity and poverty levels, confirms current literature which finds socio-economic factors are risk factors for HIV and other health conditions, operating through complex pathways [12, 39, 56–58]. Challenges in securing stable and safe housing, unemployment, and transportation that are associated with poverty, have cumulative effects that impede retention in care [31, 34, 59]. Research and large-scale implementation of macro-level, structural interventions, including education; universal healthcare; policies designed to reduce income inequality and increase income of the poor such as progressive income tax and increases in minimum wage; and immigration policies, are needed to address some of the underlying root causes of health disparities [60–62].

There are several limitations in this study. Prevalence rates in counties are not adjusted for age, and given the large difference in median age for Hispanics/Latinos (28.7) and NL-White (43.5) [63], this age difference could explain some of the observed disparities, even after adjustment for county-level variables. To examine this more carefully, we compared the national prevalence rates of HIV for Latinos and NL-Whites by age group; such data are only available at the national level [1]. The overall PRR among the entire US, treating age by race/ethnicity as two correlated random effects, comparable to our county-level analysis, was equal to 3.3 (and an age-indirect adjusted PRR calculated similar to a standardized mortality ratio of 3.10). Thus the national level age-adjusted analysis does not diminish the disparity effect and is in fact 30% higher than the PRR of the 780 counties in our generalized mixed effect analyses (PRR = 2.54). In S2 Technical Appendix Table 1 in S1 Text, we show the age specific PRRs for each of the following age groups: 13–24, 25–34, 35–44, 45–55, and 55+. Each age specific PRR exceeded or was equal to the unadjusted PRR, and was actually greatest for the eldest age group, which is disproportionately NL-White. That same table provides age-specific correlations between HIV prevalence and county size that would be required to fit HIV disparities if age were a confounder (see S2 Technical Appendix Section V in S1 Text). We found that these age-specific correlations between county prevalence and size would vary from near zero for the youngest group to 0.19 for those aged 45–54. Because of these large variations and the large PRR for each age group in the national data, we are reasonably confident that a county-level adjustment by age would not explain the Latino disparities in our county-level data. However, without having access to county-level HIV prevalence by age, we cannot completely rule out county-level age confounding of disparities due to Simpson's paradox [64], in which a positive relationship could appear in the marginal relationship between ethnicity and prevalence but be absent when conditioning on another variable, (e.g., age).

Another limitation involves known HIV status. The prevalence data used in our analysis represents people living with diagnosed HIV infection yet there are significant race/ethnicity and age differences in the percent of people who have HIV but have not been diagnosed which may result in our findings under-estimating Latino disparities. CDC's latest estimates show that the percent of Latinos living with undiagnosed HIV infection is 45% higher than for NL Whites (16.7% vs. 11.5%) and both Latino and NL Whites between the ages of 13–24 have a significantly higher percentage of undiagnosed infection (49% and 42% respectively) than those 25 and older (16% and 11% respectively) [65].

We also note that our set of county-level predictors is incomplete. Even with the availability of ACS measures, there were few race/ethnicity specific health and access to care variables at the county-level. It is likely that other indices involving direct comparisons of Latinos to NL-Whites, similar to the relative poverty prevalence that we did examine, would explain more of the disparities. While this paper includes information on percent distribution of

various Latino origin subgroups (e.g., Mexican, Puerto Rican), the differences in country of origin, whether or not they are US born, historical and cultural experiences, acculturation, and regional variation among Latinos, are not considered in our analyses. Our analyses can thus examine shared aspects of the Latino experience in the US, but it contributes relatively little to unique aspects in this diverse population group.

Despite these limitations, our findings of Latino disparities across 94% of counties and significant findings of factors that have modest explanatory power point to a pervasive health equity challenge. While much research has examined individual level risk factors as explanations for disparities, our ecological analyses suggest that network, community, and societal levels factors continue to play major roles in driving Latino disparities. Fortunately, five decades of research has finally achieved effective biomedical interventions that can greatly reduce transmission of HIV. The use of HIV medicines to achieve viral suppression among people living with HIV and prevent infection among people at increased risk of HIV acquisition, are central to the national *Ending the HIV Epidemic* strategy [66]. However, in order to overcome the nearly universal disparities faced by Latinos, we must develop, adapt and implement both micro- and macro-level interventions that are more culturally and geographically targeted prevention and care services to increase availability, initiation, and retention of PrEP and ART. Improvements in both the delivery systems for these medications and in reduction of the extensive societal barriers faced by Latinos are needed to make these medications acceptable as well as affordable and readily available.

## Supporting information

**S1 Table. Description and data sources for county-level factors.**
(PDF)

**S1 Text. Technical appendix.**
(PDF)

## Acknowledgments

We thank Dr. Jessica Jakubowski for her comments on early versions of this paper.

## Author Contributions

**Conceptualization:** Nanette D. Benbow, C. Hendricks Brown.

**Data curation:** David A. Aaby.

**Formal analysis:** David A. Aaby, Eli S. Rosenberg, C. Hendricks Brown.

**Investigation:** Nanette D. Benbow.

**Methodology:** David A. Aaby, C. Hendricks Brown.

**Project administration:** Nanette D. Benbow, C. Hendricks Brown.

**Validation:** C. Hendricks Brown.

**Visualization:** David A. Aaby.

**Writing – original draft:** Nanette D. Benbow, David A. Aaby, C. Hendricks Brown.

**Writing – review & editing:** Nanette D. Benbow, David A. Aaby, Eli S. Rosenberg, C. Hendricks Brown.

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
