## [Decision Letter · Decision Letter 0]

3 Jun 2020

PONE-D-20-10766

County-level factors affecting Latino HIV disparities in the United States

PLOS ONE

Dear Dr. Benbow,

Thank you for submitting your manuscript to PLOS ONE. After careful consideration, we feel that it has merit but does not fully meet PLOS ONE’s publication criteria as it currently stands. Therefore, we invite you to submit a revised version of the manuscript that addresses the points raised during the review process.

We look forward to receiving your revised manuscript.

Kind regards,

Claudia Marotta

Academic Editor

PLOS ONE

Journal Requirements:

Reviewers' comments:

Reviewer's Responses to Questions

**Comments to the Author**

1. Is the manuscript technically sound, and do the data support the conclusions?

Reviewer #1: Yes

Reviewer #2: Yes

2. Has the statistical analysis been performed appropriately and rigorously? 

Reviewer #1: Yes

Reviewer #2: Yes

3. Have the authors made all data underlying the findings in their manuscript fully available?

Reviewer #1: Yes

Reviewer #2: Yes

4. Is the manuscript presented in an intelligible fashion and written in standard English?

Reviewer #1: Yes

Reviewer #2: Yes

5. Review Comments to the Author

Reviewer #1: Please see attached comments.

Reviewer #2: The authors wrote a very interesting paper on the determinants of HIV disparities in a specific HIV population.

I suggest to publish it after some minor revisions

Only some suggestions to improve the manuscript:

1. reorganizes according to the editorial guidelines of the journal

1. Introduction: well done and clear, improve your introduction with aspect discussed in this article (Grabovac I, Veronese N, Stefanac S, et al. Human Immunodeficiency Virus Infection and Diverse Physical Health Outcomes: An Umbrella Review of Meta-analyses of Observational Studies. Clin Infect Dis. 2020;70(9):1809‐1815.)

2. Methods: no comments

3. Results: no suggestions

4. Discussion:

- improve your discussion with data on how low socioeconomic levels are risk factors for HIV and other infectious diseases (Di Gennaro F, et al. Prevalence and Predictors of Malaria in Human Immunodeficiency Virus-Infected Patients in Beira, Mozambique. Int J Environ Res Public Health. 2018;15(9):2032. Published 2018 Sep 17)

- and how the sharing of knowledge is the base of task shifting and has a central role to change health in more vulnerable people. (Marotta C, et al. Pathways of care for HIV infected children in Beira, Mozambique: pre-post intervention study to assess the impact of task shifting. BMC Public Health. 2018;18(1):703. Published 2018 Jun 7.)

6. PLOS authors have the option to publish the peer review history of their article (what does this mean?). If published, this will include your full peer review and any attached files.

Reviewer #1: No

Reviewer #2: Yes: Francesco Di Gennaro

---

## [Author Response · Author response to Decision Letter 0]

15 Jul 2020

Editor

1. Please ensure that your manuscript meets PLOS ONE style requirements, including those for file naming.

Based on PLOS ONE style requirements, we have made the following changes: reformatted tables, modified font size to reflect Level 3 headings, deleted “Funding” and “Author Contribution”, reformatted figures, and renamed supporting documents files. We also updated references using PLOS ONE reference guidance described in https://journals.plos.org/plosone/s/submission-guidelines#loc-references.

Reviewer 1

The manuscript is interesting and could be accepted after revision.

Authors should improve the discussion, better explaining the lesson learnt and possible solutions to this problem.

Specifically:

1) Reviewer comment: Authors stated: “Increasing the availability, initiation, and retention of PrEP and ART are essential strategies to eliminate HIV.”….It has to be better contextualized and explained. In which way? In fact, in the previous paragraph they stated, “In such settings, language and cultural differences could be significant barriers to HIV testing and treatment that would differentially affect Latinos across both HIV prevention and treatment cascades”…How authors think possible increase PrEP and ART? See ref: DOI: 10.1186/s12889-018-5646-8 

Response: To address this and other comments by both reviewers, we restructured the discussion section completely. We addressed this comment in lines 507-523 (in track-changes version) with changes, additions, and new references that provide context and potential solutions to engagement and retention to HIV care services:

2) Reviewer comment: Authors should better explain also the following: “We conjecture that focusing primarily on changing individual level risk behavior by itself is not likely to overcome all these disparities” how much important it this for authors? 

Response: In discussing this conjecture, we restructure the discussion to highlight how our findings are aligned with the literature that describes the multiple levels contributing to disparities, following a socioecological framework of health. In lines 507-523, we discuss individual and provider level barriers to access to care that can lead to disparities and identify interventions that can be adapted to address them. In lines 524-544, we present community-level factors and suggest approaches and interventions that may be tailored for Latinos to improve access and use to needed services. In lines 553-572, we discuss individual level factors associated with injection drug use. In lines 577-586, we present socioeconomic, societal level factors strongly associated with HIV and other health disparities among Latinos. 

3) Reviewer comment: Authors should create a priorities list of actions to tackle this problem.

Response: We agree and this comment guided the way we restructured the Discussion section. These proposed actions and approaches address individual, community, and societal levels as described in item 2 above. 

4) Reviewer comment: Social context was mentioned many time but never well discussed, maybe authors could consider these references: https://doi.org/10.1016/j.tube.2017.01.002; doi: 10.1177/00333549101250S405

Response: We appreciate this observation and it also informed restructuring of the Discussion section. We use the socioecological framework of health in which social context is a key element. To address this specifically, we added lines 524-544 that address existing research on successful outreach approaches to Latinos and new research that is needed. 

Reviewer 2

The authors wrote a very interesting paper on the determinants of HIV disparities in a specific HIV population.

I suggest to publish it after some minor revisions

Only some suggestions to improve the manuscript:

1) Reviewer comment: reorganizes according to the editorial guidelines of the journal

Response: We thank the reviewer for pointing this out. Changes have been made and are described in item #1 in response to Editor comment.

2) Reviewer comment: Introduction: well done and clear, improve your introduction with aspect discussed in this article (Grabovac I, Veronese N, Stefanac S, et al. Human Immunodeficiency Virus Infection and Diverse Physical Health Outcomes: An Umbrella Review of Meta-analyses of Observational Studies. Clin Infect Dis. 2020;70(9):1809‐1815.)

 Response: We agree with suggestion and appreciate the reference. We provided additional context by describing the role of the HIV continuum of care and reaching viral suppression. In lines 54-60 and 73-76.

4) Discussion:

Reviewer comment: improve your discussion with data on how low socioeconomic levels are risk factors for HIV and other infectious diseases (Di Gennaro F, et al. Prevalence and Predictors of Malaria in Human Immunodeficiency Virus-Infected Patients in Beira, Mozambique. Int J Environ Res Public Health. 2018;15(9):2032. Published 2018 Sep 17)

Response: We appreciate the reference and incorporated it into our revision found in lines 577-586 

Reviewer comment: and how the sharing of knowledge is the base of task shifting and has a central role to change health in more vulnerable people. (Marotta C, et al. Pathways of care for HIV infected children in Beira, Mozambique: pre-post intervention study to assess the impact of task shifting. BMC Public Health. 2018;18(1):703. Published 2018 Jun 7.)

Response: While task shifting is a potentially effective approach to address the human resource limitations and to scale up the ART coverage and retention in care, we could not find any evidence of the role of task shifting to address specific barriers and facilitators faced by Latinos in the US. However, we did identify evidence of the impact that culturally appropriate HIV care can have on retention in care, and delivering care via telemedicine to reach Latinos in areas where PrEP and HIV care may not be readily available. This is described in lines 507-523.

---

## [Decision Letter · Decision Letter 1]

23 Jul 2020

County-level factors affecting Latino HIV disparities in the United States

PONE-D-20-10766R1

Dear Dr. Nanette Benbow,

We’re pleased to inform you that your manuscript has been judged scientifically suitable for publication and will be formally accepted for publication once it meets all outstanding technical requirements.

Kind regards,

Claudia Marotta

Academic Editor

PLOS ONE

Additional Editor Comments (optional):

Dear Authors,

I appreciated the good interaction between reviewers and authors that according to me increased the quality of the manuscript that is now ready to be accepted for publication.

Best Regards and congratulations

Reviewers' comments:

Reviewer's Responses to Questions

**Comments to the Author**

1. If the authors have adequately addressed your comments raised in a previous round of review and you feel that this manuscript is now acceptable for publication, you may indicate that here to bypass the “Comments to the Author” section, enter your conflict of interest statement in the “Confidential to Editor” section, and submit your "Accept" recommendation.

Reviewer #1: All comments have been addressed

Reviewer #2: All comments have been addressed

2. Is the manuscript technically sound, and do the data support the conclusions?

Reviewer #1: Yes

Reviewer #2: Yes

3. Has the statistical analysis been performed appropriately and rigorously? 

Reviewer #1: Yes

Reviewer #2: Yes

4. Have the authors made all data underlying the findings in their manuscript fully available?

Reviewer #1: Yes

Reviewer #2: Yes

5. Is the manuscript presented in an intelligible fashion and written in standard English?

Reviewer #1: Yes

Reviewer #2: Yes

6. Review Comments to the Author

Reviewer #1: (No Response)

Reviewer #2: Authors improved their manuscript and I appreciaet a lot this new version

I think that manuscript can be accept in this present form.

congratulations

7. PLOS authors have the option to publish the peer review history of their article (what does this mean?). If published, this will include your full peer review and any attached files.

Reviewer #1: No

Reviewer #2: **Yes: **Francesco Di Gennaro

---

## [Editor Report · Acceptance letter]

28 Jul 2020

PONE-D-20-10766R1 

County-level factors affecting Latino HIV disparities in the United States 

Dear Dr. Benbow:

I'm pleased to inform you that your manuscript has been deemed suitable for publication in PLOS ONE. Congratulations! Your manuscript is now with our production department. 

Kind regards, 

on behalf of

Dr. Claudia Marotta 

Academic Editor

PLOS ONE